# New and Preliminary Evidence on Altered Oral and Gut Microbiota in Individuals with Autism Spectrum Disorder (ASD): Implications for ASD Diagnosis and Subtyping Based on Microbial Biomarkers

**DOI:** 10.3390/nu11092128

**Published:** 2019-09-06

**Authors:** Xuejun Kong, Jun Liu, Murat Cetinbas, Ruslan Sadreyev, Madelyn Koh, Hui Huang, Adetaye Adeseye, Puhan He, Junli Zhu, Hugh Russell, Clara Hobbie, Kevi Liu, Andrew B. Onderdonk

**Affiliations:** 1Athinoula A. Martinos Center for Biomedical Imaging, Massachusetts General Hospital, 149 13th Street Charlestown, Boston, MA 02129, USA; 2Harvard Medical School, Boston, 25 Shattuck Street, Boston, MA 02115, USA; 3Department of Molecular Biology, Massachusetts General Hospital, 185 Cambridge Street, Boston, MA 02114, USA; 4Harvard School of Dental Medicine, 188 Longwood Avenue, Boston, MA 02115, USA; 5Fisher college, 1 Arlington Street, Boston, MA 02116, USA; 6Precidiag, Inc., 27 Strathmore Road, Natick, MA 01760, USA; 7Brigham and Women’s Hospital, 75 Francis Street, Boston, MA 02115, USA

**Keywords:** autism spectrum disorders, gut microbiota, oral microbiota, dysbiosis, co-occurring conditions, allergy, abdominal pain, biomarker discovery

## Abstract

Autism Spectrum Disorder (ASD) is a complex neurological and developmental disorder characterized by behavioral and social impairments as well as multiple co-occurring conditions, such as gastrointestinal abnormalities, dental/periodontal diseases, and allergies. The etiology of ASD likely involves interaction between genetic and environmental factors. Recent studies suggest that oral and gut microbiome play important roles in the pathogenesis of inflammation, immune dysfunction, and disruption of the gut–brain axis, which may contribute to ASD pathophysiology. The majority of previous studies used unrelated neurotypical individuals as controls, and they focused on the gut microbiome, with little attention paid to the oral flora. In this pilot study, we used a first degree-relative matched design combined with high fidelity 16S rRNA (ribosomal RNA) gene amplicon sequencing in order to characterize the oral and gut microbiotas of patients with ASD compared to neurotypical individuals, and explored the utility of microbiome markers for ASD diagnosis and subtyping of clinical comorbid conditions. Additionally, we aimed to develop microbiome biomarkers to monitor responses to a subsequent clinical trial using probiotics supplementation. We identified distinct features of gut and salivary microbiota that differed between ASD patients and neurotypical controls. We next explored the utility of some differentially enriched markers for ASD diagnosis and examined the association between the oral and gut microbiomes using network analysis. Due to the tremendous clinical heterogeneity of the ASD population, we explored the relationship between microbiome and clinical indices as an attempt to extract microbiome signatures assocociated with clinical subtypes, including allergies, abdominal pain, and abnormal dietary habits. The diagnosis of ASD currently relies on psychological testing with potentially high subjectivity. Given the emerging role that the oral and gut microbiome plays in systemic diseases, our study will provide preliminary evidence for developing microbial markers that can be used to diagnose or guide treatment of ASD and comorbid conditions. These preliminary results also serve as a starting point to test whether altering the oral and gut microbiome could improve co-morbid conditions in patients with ASD and further modify the core symptoms of ASD.

## 1. Introduction

Autism Spectrum Disorder (ASD) is a complex neurological and developmental disorder with a rapidly increasing prevalence on a global scale [1]. The etiology of ASD likely involves an interplay between genetic and environmental factors, as well as both systemic inflammation and inflammation of the central nervous system (CNS) [2,3,4]. Recent studies suggest that microbiome dysregulation plays an important role in the pathogenesis of inflammation [5,6,7,8], which may contribute to the manifestation of ASD symptoms [9,10,11,12]. Evidence from animal studies supports a link between microbiome dysregulation, inflammation in the body, and development of ASD [13,14]. Patients with autism often have difficulties maintaining a balanced diet, due to multiple factors such as highly selective food preference, organic gastrointestinal (GI) diseases, and oral motor difficulties, and they show high rates of gut dysbiosis compared to neurotypical individuals [9,12]. Notably, some studies demonstrated a correlation between the severity of GI dysfunction and the severity of behavioral symptoms [15]. Gut dysbiosis may affect the CNS via the vagus nerve, microbial metabolites and neuroinflammation [16,17,18].

While most studies agree that the microbiome composition is different between autistic and neurotypical populations, these studies have yielded inconsistent results as to the nature or extent of these GI bacterial community differences [12,19]. Environmental factors are the dominant determinants for gut microbiome composition [20,21,22], yet most previous studies using age and sex matched controls have not adequately controlled for environmental influences [12,23,24]. In addition, compared to the gut, the oral microbiome is understudied, despite dental plaque and saliva samples being easier to obtain than stool samples. Alterations of the oral microbiota are associated with not only periodontal diseases [25], but also the upper GI tract flora [26], systemic diseases such as Rheumatoid Arthritis [27] and neurological conditions such as Alzheimer’s disease [28]. Epidemiological studies have demonstrated a higher prevalence of oral health issues among patients with ASD, as compared to neurotypical individuals [29]. Only two studies to date have explored differences in oral microbiota between children with autism and controls [30,31]. Results from these studies have low degrees of concordance, likely due to the different sequencing methodologies and study designs.

Here, we have designed a pilot study to investigate the oral and gut microbiome simultaneously in patients with ASD and their first-degree family members. This would control for genetic and lifestyle factors while investigating the existence of ASD-microbiome signatures and whether these signatures hold any diagnostic value. Furthermore, to explore the poorly understood oral microbiome, we have directly compared oral and gut microbiome to explore their relationship in ASD and their association with systemic clinical indices. These questions are important to address in order to detail the roles of the human microbiome in ASD, and its utility in guiding diagnosis of ASD, clinical subtypes, and potential targeted interventions.

Given the multitude of factors that influence microbiome-host interactions, a secondary goal of the study attempts to characterize the potential relationships between the gut and oral microbiome and relevant clinical indices, including allergy, abdominal pain and dietary habits. Previously, Plaza-Diaz investigated gut microbiome in ASD patients with or without mental regression and found microbiome signatures associated with different psychiatric subtypes [32]. However, the association between medical subtypes and microbiome has been poorly explored in ASD patients.

Research on high impact diseases such as Rheumatoid Arthritis has revealed fascinating associations between oral and gut microbiomes [27]. Our study will serve as a starting point to address the complex interplay between the oral microbiome and the gut microbiome in the phenotypic presentation and pathophysiology of ASD. We believe that this study will open new horizons and opportunities in disease investigation and management. As a pre-probiotics clinical trial pilot project, we hope that this study and its continuation will provide insight for whether this new methodology with combined oral and fecal data can be used to (1) screen, diagnose, and determine subtypes of ASD, (2) stratify patients who may respond to probiotics therapy, (3) provide guidance on treatment strategies and develop targeted probiotic formulation, and (4) help to monitor treatment efficacy.

## 2. Materials and Methods

### 2.1. Study Participants

We recruited 20 patients diagnosed with ASD (autism spectrum disorder) and compared them with 19 family members (parent or sibling) as neurotypical controls. Patients had been diagnosed with ASD according to DSM-5 (Diagnostic and Statistical Manual of Mental Disorders) criteria [33]. Individuals with ASD between 7–25 years old with a disease duration of at least 6 weeks were enlisted. Exclusion criteria for all subjects included known genetic conditions, clinically evident serious infections or inflammatory conditions, history of cancer, severe dental/periodontal diseases or possession of dental braces. Subjects who had received probiotic treatment were asked to stop treatment at least one week prior to sample collection and subjects were excluded if they had taken antibiotics in the preceding month. Neurotypical controls had to meet the following criteria: biological sibling or biological parent of autistic subjects with IQ equal to or greater than 80 who do not have a diagnosis of ASD, attention deficit hyperactivity disorder, other intellectual developmental disorders, or psychiatric conditions. For recruitment of control subjects, siblings of the same gender and comparable age (+/− 5 years apart) received the highest priority, but an opposite-gender sibling was recruited for a control as needed. If the subject with ASD had no siblings, a parent acting as primary caretaker was recruited. Demographics and characteristics of study subjects are available in Appendix A and summarized in Table 1. Visual dental inspections were performed to determine oral health status for all subjects. Lifestyle questionnaires were distributed to assess factors that could affect microbiome status and create a GI clinical indices (GSI) score (Table 1, Appendix A) [34,35].

ASD patients were recruited from clinics at Massachusetts General Hospital (MGH), Beth Israel Deaconess Medical Center, community ASD education events, and charity ASD programs in Boston. The study was approved by institutional review board of MGH (Boston, MA, USA, IRB protocol number: 2017P000573). Informed consents were obtained from subjects or the legal guardians of the subjects. All methods were performed in accordance with the relevant guidelines and regulations.

### 2.2. Sample Handling and Collection

To obtain oral microbiome samples, participants were asked to produce 1–3 mL of saliva after refraining from eating, drinking and oral hygiene practice for 1 h. Samples were collected with sterile DNA- and RNA-free 15 mL Falcon tubes and immediately frozen at −80 °C. De-identified and coded samples were shipped to Precidiag Inc. (Natick, MA, USA) for DNA extraction and sequencing on dry ice. Stool samples were collected by the participants at home under the supervision of trained parents with a HR-Easy Stool Collection Kit (Precidiag, Inc.) and stored at room temperature, followed by de-identification and shipment to a Precidiag CLIA-certified laboratory for DNA extraction and sequencing analysis. The HR-Easy Stool Collection Kit provides a superior method for collection, storage and stabilizing stool samples for microbiome study at ambient temperature for up to a month with minimal alterations when compared with freshly-collected samples (Yu et al., manuscript in preparation). Microbial DNA was then extracted using a HR-Easy Fecal DNA Kit (Precidiag, Inc.) according to the manufacturer’s instructions and DNA samples were carefully quantified with a nanodrop spectrophotometer. A260/A280 ratios were also measured to confirm high-purity DNA yield. DNA samples were frozen at −20 °C until use.

### 2.3. 16S rRNA Gene Amplicon Sequencing

Microbial 16S rRNA V3-V4 genomic regions from total oral and gut DNA samples were amplified with the following primers 341F: 5′AATGATACGGCGACCACCGAGATCTA-CACTCTTTCCCTACACGACGCTCTTCCGATCTCCTACGGGAGGCAGCAGCCTACGGGNBGCASCAG3′ and 805R: 5′CAAGCAGAAGACGGCATACGAGATNNNNNNG-TGACTGGAGTTCAGACGTGTGCTCTTTCCGATCTGACTACNVGGGTATCTAATCC3′ via polymerase chain reaction (PCR) (95  °C for 2 min, followed by 25 cycles at 95 °C for 30 s, 55 °C for 30 s, and 72 °C for 30 s, and a final extension at 72 °C for 5 min). PCR products were purified and analyzed using a Bioanalyzer DNA kit, followed by quantification with real-time PCR. Serially diluted PhiX control library (Illumina, San Diego, CA, USA) was included as a standard. DNA libraries were pooled and sequenced on an Illumina MiSeq next-generation sequencing system (Illumina; CA) using a V2 2 × 250 bp paired-end protocol with overlapping reads.

Of note, we included strict quality control processes involving microbial DNA extraction, 16S rRNA gene amplicon amplification, and amplicon sequencing with a set of controls that enabled us to evaluate the potential introduction of contaminants or off-target amplification. Non-template controls (extraction chemistries) were included in the microbial DNA extraction process and the resulting material was subsequently used for PCR amplification. Additionally, at the step of amplification, another set of non-template controls (PCR-mix) was included to evaluate the potential introduction of contamination at this step. Similarly, a positive control comprised of known and previously characterized microbial DNA was included at this step to evaluate the efficiency of the amplification process. Before samples were pooled together, sequencing controls were evaluated, and samples were rejected if the presence of amplicons in any of the non-template controls or the absence of amplicons in the positive control was detected. In the present study, no amplicons were observed in the non-template controls and a negligible number of raw reads were recovered after sequencing.

### 2.4. Sequencing Data Processing

Sequencing data were processed and analyzed with a QIIME software package v. 2018.2.0 [36]. The sequencing reads with a low quality score (average Q < 25) were truncated to 240 bp, followed by filtering using the deblur algorithm with default settings [37]. The remaining high-quality reads were aligned with the reference library using mafft [38]. Next, the aligned reads were masked to remove highly variable positions, and a phylogenetic tree was generated from the masked alignment using the FastTree method [39]. Taxonomy assignment was performed using the feature-classifier method and naïve Bayes classifier trained on the Greengenes 13_8 99% operational taxonomic units (OTUs) (Appendix A).

### 2.5. Biostatistical Analysis

#### 2.5.1. Variables Measured

The main variables are the compositions of oral and gut microbiome, and quantities of microbes on genus and phylum level within each sample (OTUs). Other variables include patients’ demographic information, baseline medical conditions, lifestyle factors and clinical indices.

#### 2.5.2. Alpha and Beta Diversity

Alpha diversity was calculated on the basis of the gene profile for each sample based on the Shannon index, Faith’s index, and Simpson’s evenness index [40,41,42]. Beta diversity was calculated on the unweighted and weighted UniFrac distances, Jaccard and the Bray–Curtis dissimilarity [43,44]. Alpha and beta-diversity estimates were computed using QIIME2 [36]. Alpha and beta diversity metrics and Principal Component Analysis plots based on the Jaccard distance were generated using default QIIME2 plugins [36,43,45,46,47].

Kruskal–Wallis tests were used to compare alpha diversity between ASD patients and controls for oral or gut microbiome respectively. A cut off false discovery rate (FDR) of 0.05 based on the Benjamini–Hochberg (BH) method was applied [48]. Comparison of beta diversity indices were calculated by Permutational multivariate analysis of variance (PERMANOVA).

#### 2.5.3. Statistical Analyses of Differentially Enriched Microbiome Taxa

Significant differences in the relative abundance of microbial genera and phyla between individuals with ASD and controls were identified by Kruskal–Wallis tests and BH adjustment for multiple comparisons. In addition, we performed a paired Wilcoxon signed-rank test on the relative abundances with BH adjustment. Furthermore, we explored differential bacteria enrichment on all taxonomy levels using the ANCOM (Analysis of Composition of Microbiomes) method, an algorithm that accounts for compositional constraints to reduce false discoveries in detecting differentially abundant taxa at an ecosystem level, while maintaining high statistical power [49]. An FDR cutoff of 0.2 was applied for taxa-level comparison [50].

### 2.6. Microbiome Biomarker Discovery

In order to measure whether the relative abundance of gut and oral microbial taxa and the dysbiosis markers could classify ASD and control groups correctly, we created a receiver operator characteristics (ROC) curve using Prism GraphPad (version 7.00 for Mac, GraphPad Software, La Jolla, San Diego, CA, USA, www.graphpad.com). Statistical significance of areas under the curves (AUCs) for dysbiosis markers were performed with the default plugin of Prism GraphPad.

### 2.7. Microbiome Network Analysis

In order to assess the taxonomic relatedness/association within the gut and oral microbiota as well as between oral and gut microbiota, we performed correlation-based network analysis using the SparCC (Sparse Correlations for Compositional data) method [51,52]. We performed SparCC for microbiome data on phylum and genus level from all subjects, as well as within ASD and control groups, respectively (Correlation coefficient cut-off = 0.3).

### 2.8. Influence of Clinical and Lifestyle Factors

Kruskal–Wallis tests with BH adjustment for FDR were used to assess differential abundance of dysbiosis markers and bacterial taxa (phylum and genus level) between binary clinical classifiers (i.e., presence or absence of allergy, constipation and abdominal pain) with a FDR cut off of 0.2. Relevant clinical indices were treated as binary even though some data were collected as ordinal (e.g., GSI scores). Analysis was further stratified by ASD and control groups. Genus level analysis was performed with genera that have a relative abundance of at least 0.5%. We compared the dietary habits between ASD patients and neurotypical controls based on numerical scores from baseline survey questions. The responses for each question were recorded on a numerical scale from 0 to 4, where a larger score indicated that the subject exhibited the behavior with greater prominence. We next assessed the correlation between eating habit scores, allergy/autoimmunity scores, GSI total score, and key ASD gut microbiome markers in patients with ASD. We used the Spearman’s correlation and an FDR cutoff of 0.05.

### 2.9. Softwares Used

QIIME software package v. 2018.2.0 [36], RStudio (RStudio Team, 2017), R (R Core Team, 2017) and Prism GraphPad version 7.00 for Mac (GraphPad Software, La Jolla, CA, USA) were used for statistical testing and graph generation. Adobe Illustrator CC was used for figure editing.

## 3. Results

To characterize the gut and oral microbiota associated with autism, we recruited 20 autistic subjects and 19 controls (Table 1). Of the controls, 8 were neurotypical biological parents and 11 were neurotypical biological siblings. Demographic information is summarized in Table 1. One family had 1 parental control with 2 ASD children. Overall, there were significant inter-subject and inter-pair variabilities in microbiota composition (Figure 1A,B, Appendix A).

### 3.1. Autistic Subjects Harbor an Altered Oral Microbiota Compared to First Degree-Family Member Controls

Consistent with previous studies, analysis of alpha diversity calculated by the Shannon index revealed no significant differences between autistic and neurotypical subjects’ salivary microbiota (Appendix A). A heatmap (Figure 1C) visiually demonstrates that the beta diversity calculated on the unweighted, weighted UniFrac distances and the Bray–Curtis dissimilarity revealed no significant difference between the ASD and control groups for oral flora (Figure 2A, Appendix A, PERMANOVA). The major phyla that contributed to the oral microbiome in ASD and control groups are summarized in Figure 2C. On the genus level, the ASD and control groups share 9 out of 10 most abundant genera, including *Prevotella*, *Fusobacterium*, *Rothia*, *Haemophilus*, *Streptococcus*, *Neisseria*, *Veillonella*, and an unknown genus in the *Neisseriaceae* family.

We found differential enrichment of bacterial taxa in the oral microbiota of autistic individuals compared to the controls. On the phylum level, ASD patients showed a trend of lower relative abundance of *TM7* bacteria (Figure 2D, Appendix A). In total, 6 genera showed altered relative abundance between the two groups (Kruskal–Wallis test, *p* < 0.05, Figure 3A, Figure 4B, Appendix A). In particular, the relative abundance of an unspecified genus in the class of *Bacilli* was statistically significant after adjusting for the false discovery rate (FDR) (Figure 5B, Appendix A). 

### 3.2. Autistic Subjects Harbor an Altered Bacterial Gut Microbiota Compared to First Degree-Family Member Controls

Consistent with previous studies, the analysis of gut alpha and beta diversity as well as principal component analysis (PCA) revealed no significant differences between autistic and neurotypical subjects (Figure 2B, PERMANOVA, Appendix A), as visualized by a heatmap (Figure 1D). On the phylum level, *Firmicutes*, *Bacteroidetes* and *Proteobacteria* are the most abundant gut phyla in both ASD patients and control subjects, comprising more than 90% of all operational taxonomic units (OTUs) (Figure 2C). On the genus level, ASD and control groups share 9 out of 10 most abundant genera, including *Bifidobacterium*, *Blautia*, *Prevotella*, *Bacteroides*, *Faecalibacterium*, and unknown genera in *Ruminococcaceae* family, *Lachnospiraceae* family, *Enterobacteriaceae* family and *Clostridiales* order.

Further analysis of the dysbiosis markers revealed differences in the gut microbiota of subjects with autism and their family member controls. Several phylum level markers showed statistically significant changes between ASD and control, including *Firmicutes*/*Bacteroidetes* ratio (Figure 3C) likely driven by *Bacteroidetes* (Figure 2E, Appendix A). The phylum *Proteobacteria* is associated with metabolic syndrome and inflammatory bowel disease (IBD), and normally makes up less than 10% of the gut microbiome in healthy individuals [53]. Among the six subjects with significant *Proteobacteria* overgrowth (with relative abundance values greater than 30%), 4 were ASD patients (Appendix A). On the genus level, 6 taxa showed trends of altered abundance between the two groups, including *Paraprevotella*, *Granulicatella, Butyricimonas*, *cc_115*, *Peptoniphilus* and *Eubacterium* (Figure 5A, Appendix A).

### 3.3. Gut and Saliva Biomarkers Can Classify ASD and Control Groups

In order to measure how correctly the relative abundance of gut and oral microbial taxa and the dysbiosis markers could classify two groups of samples, we created a receiver operator characteristics (ROC) curve, which is a common methodology used to evaluate classification performance of potential biomarkers (Figure 3D). The performance of a potential classifier (binary) can be evaluated by measuring the area under the curve (AUC), which represents true versus false positive rates. An AUC value of 0.5 corresponds to random classification and a value of 1.0 corresponds to perfect classification. Taking all gut and saliva genera as well as gut dysbiosis markers that showed statistically significant differential expression (Kruskal–Wallis tests) from previous analyses, two genera (gut *Butyricimonas*, saliva *Parvimonas*, Figure 3A,B) and the well-recognized dysbiosis marker gut *Firmicutes*/*Bacteroidetes* ratio (Figure 3C), all showed the highest AUC values (up to 0.724) with *p* value < 0.05 (Figure 3D, Appendix A).

### 3.4. Results of Paired Analysis Overlap Partially with Group Analysis

Due to the nature of paired study design, we also performed paired a Wilcoxon signed-rank test on the relative abundance of the OTUs, in addition to Kruskal–Wallis tests, by subject groups (ASD vs. control). Those with significant Wilcoxon’s *p* values had partial overlap with results from grouped Kruskal–Wallis tests (Figure 4A,B). However, after adjustment for multiple comparison, FDRs from paired analyses were not statistically significant (Appendix A). Examples of gut and oral genera that showed the most significant pairwise changes are recorded in Figure 4C–F. Due to high inter-individual variabilities, subsequent analysis consisted of group-wise approaches.

In addition to Kruskal–Wallis tests with FDR adjustment, we explored differential bacteria enrichment on all taxonomy levels using the more conservative ANCOM method [49]. This method did not reveal statistically significant differences in the enrichment patterns detected by the Kruskal–Wallis test (Appendix A). 

### 3.5. Exploring the Relationship between Gut–Oral Microbiome and Their Co-Occurrence Network

Since the current project characterized gut and oral microbiota samples from the same subjects, we explored the relationship between gut and oral microbiota within individuals. Consistent with previous publications, we found that the gut and oral microbiome are distinct, based on beta diversity indices and PCA (Figure 5B, PERMANOVA). This can be seen through heatmap clustering (Figure 5A) as well as the OTU level ANCOM analysis (Appendix A).

In order to assess the taxonomic association within the gut and oral microbiota as well as between oral and gut microbiota in a non-biased manner, we performed correlation-based network analysis using the Sparse Correlations for Compositional data (SparCC) method [51,52] (Figure 5C). This method is capable of estimating correlation values from compositional data and has been validated as a superior analysis technique than Pearson’s correlation methods for compositional data such as 16S rRNA gene amplicon sequencing [51]. The goal of this analysis is to infer any potential synergistic relationships between bacterial taxa within a community and between communities. We also hoped to detect GI dysbiosis purely using salivary microbial markers because, due to high prevalence of constipation in the ASD population, it is much easier to obtain saliva samples than stool samples. The salivary microbiome could then serve as a diagnostic window into the GI environment of the ASD patients. Previously, network correlation analysis has yielded important insights regarding bacterial community structures related to enterotypes [54].

Overall, the oral microbiome exhibits a denser co-occurrence network compared to the gut, both at the phylum and genus level (Figure 5C–E). The same trend holds true when analyzing ASD subjects and control subjects separately (Figure 5D,E). Within the salivary co-occurrence network at the phylum level, the highest correlations are observed in a cluster consisting of *Actinobacteria*, *Proteobacteria*, *Firmicutes* and *Bacteroidetes* (Figure 5C, dotted circle), especially between *Firmicutes* and *Actinobacteria* (Appendix A). Importantly, some gut and oral phylum show positive inter-community co-occurrence. There is a positive correlation between saliva *Verrucomicrobia* and gut *Actinobacteria* (Figure 5C). In the ASD population but not the controls, gut *Firmicutes*, which is a known dysbiosis marker, showed positive correlation with saliva level of *Chloroflexi* (Figure 5D,E). We then computed the co-occurrence network on the genus level using bacteria genera that make up at least 0.5% of all OTUs. The genus-level co-occurrence density was notably higher compared to phylum level (Appendix A), as many genera demonstrated intra-community co-occurring relationships. In terms of inter-community co-occurrence, several gut genera, including *Bifidobacteria*, *Dialister*, *Escherichia*, *SMB53* and an unspecified genus in *Enterobacteriaceae* all exhibited positive correlation with salivary genera in the control subjects, whereas only *Escherichia* and an unspecified genus of *Clostridiales* showed co-occurrence with saliva genera in ASD patients (Appendix A).

Alpha diversity has been conventionally used as an index for dysbiosis, as low alpha diversity indicates diminished community richness and potentially diminished resilience to disturbances. Alpha diversity shows a positive correlation between the gut and oral microbiota, although it is not statistically significant (Appendix A).

### 3.6. Microbiome Signatures in Clinical Subtypes

Due to the tremendous clinical heterogeneity of the ASD population, we explored the relationship between microbiome and clinical indices as an attempt to extract microbiome signatures assocociated with clinical subtypes. We focused on three major medical comorbidities that have previously reported associations with microbiome, including allergy, GI disturbances and poor diet.

### 3.7. Allergies

We first investigated whether phylum level dysbiosis markers (including gut *Proteobacteria*, *Firmicutes*, and *Bacteroidetes*, and oral *SR1* and *Synergistetes*) may be associated with disease states. Among all clinical indices assessed, the incidences of allergy were notably higher in the ASD group (7/19 vs. 11/18, Chi-square test, *p* value < 0.05). The relative abundance of oral *SR1* is significantly lower in ASD patients who also have allergies, in comparison to ASD patients without allergies, but this trend is not present in control subjects (Kruskal–Wallis, Figure 6A). Subjects with allergies also showed increased relative abundance of gut *Proteobacteria*, a phylum previously associated with autoimmune conditions (Kruskal–Wallis, Figure 6B). These differences are detected only in ASD patients and not in controls (Figure 6B). All ASD subjects who had significant gut *Proteobacteria* overgrowth (>30%) also suffered from allergies (4/4), whereas none of the 2 control subjects with *Proteobacteria* overgrowth did (0/2).

We next performed genus level correlation analysis of the oral and gut bacterial relative abundances against allergy status, using bacteria genera that make up at least 0.5% of all OTUs. No salivary or gut genus was significantly and differentially enriched by allergy status, after stratifying by ASD and control group (Appendix A).

### 3.8. GI Disturbances

Patients with autism suffer from many co-occurring GI conditions [55]. Previous studies found that gut microbiome is associated with and may play important roles in GI symptoms such as constipation and abdominal pain [23]. We performed genus level correlation analysis of the gut bacterial relative abundances by constipation and abdominal pain status, using gut genera that make up at least 0.5% of all OTUs. *Roseburia* and *Bacteroides* were differentially enriched in subjects without abdominal pain (Figure 7A, Kruskal–Wallis, Appendix A), and this difference in *Roseburia* remained statistically significant after FDR adjustment (pain 2.7% vs. no pain 5.7%). After stratifying by ASD and control subjects, ASD patients without abdominal pain had significantly higher levels of *Bacteroides*, as compared to ASD patients with abdominal pain, whereas control subjects without abdominal pain had lower levels of *Bacteroides*, as compared to control subjects with abdominal pain (Figure 7A, Kruskal–Wallis, Appendix A).

Given the concordance between the oral microbiome and upper GI microbiome [26], it is possible that the oral microbiome may be associated with upper GI health and contribute to abdominal pain. We explored phylum and genus levels correlation analysis of the oral bacterial relative abundances between subjects with or without abdominal pain. No oral phylum showed differential enrichment, but several oral genera are differentially enriched based on abdominal pain status, including *Porphyromonas*, *Megasphaera*, *Haemophilus* (Figure 7B, Kruskal–Wallis test, Appendix A). Remarkably, *Porphyromonas* is significantly less abundant in subjects without abdominal pain after FDR adjustment (pain 0.7% vs. no pain 2.2%). When stratifying based on ASD status, ASD patients with abdominal pain showed a higher trend of *Actinomyces*, as compared to ASD patients without abdominal pain (Figure 7B, Kruskal–Wallis test, Appendix A).

The gut alpha diversity showed no difference between the constipated and non-constipated group (Figure 6C). When stratifying patients with ASD from the control group, there was an increased trend of gut alpha diversity in constipated ASD patients but not in constipated controls (Figure 6C’,C”), consistent with a previous study showing increased gut alpha diversity in functional constipation patients [56].

### 3.9. Dietary Habits and Gut Microbiome Markers

Previous studies indicate dietary challenges in ASD patients, but the association between altered dietary patterns with gut dysbiosis has not been explored in ASD patients. We found that ASD patients exhibit a statistically more restricted diet, while finding it more difficult to accept certain foods and try new foods (Mann–Whitney U test, Figure 8A–C). However, no significant differences were found between groups in respect to the amount, rate, interest, environment, or multitasking habits while eating.

We next assessed correlation between eating habit scores, allergy/autoimmunity scores, GSI total score, and key ASD gut microbiome markers in patients with ASD. Examined gut microbiome markers include Shannon alpha diversity index, gut *Firmicutes/Bacteroidetes* ratio and relative abundances of gut butyricimonas, paraprevotella, granulicatella, eubacterium, and cc_115 genera which showed significant difference between ASD and control groups based on previous grouped or paired analysis (Figure 4). Most notably, we found that ASD individuals uniquely display correlations between gut butyricimonas relative abundance, eating habit total score, and allergy/immune functions (Figure 8D). *Firmicutes/Bacteroidetes* ratio is negatively correlated with allergy/immune function while the same trends are not observed in neurotypical controls. Assessed variables lacking significant correlations with gut microbiome markers are not shown.

## 4. Discussion

In this cross-sectional study, we conducted a comparative analysis between the gut and oral microbiota of ASD children and that of healthy, first-degree relative co-inhabitant controls. Our study is the first to use a first-degree relative matched subject design combined with high fidelity next generation sequencing technology to investigate the microbiome of ASD individuals. We believe that this study design better controls for variations in genetic background and environmental factors, and therefore has better specificity for detecting ASD-related microbial signatures [23,24]. This paired control scheme has been increasingly used in microbiome studies for diseases that have strong genetic and environmental contributing factors, such as IBD [57].

Our analysis detected differences between ASD and control subjects in both their gut and oral microbiomes. We identified an unspecified oral *Bacilli* genus, the relative abundance of which is significantly different between the ASD and control groups (FDR < 0.05), which has not been described by previous reports [30,31]. Parallel to this observation, amounts of bacteria in the class *Bacilli* were significantly higher in the gut of ASD individuals compared to controls (0.7% vs. 0.4%, Kruskal–Wallis test, *p* < 0.05), consistent with findings of Adams et al. [15]. Previous studies of the gut microbiome have revealed significant increases in facultative anaerobic commensal bacteria belonging to the class *Bacilli* seen in individuals with IBD, supporting a potential connection between *Bacilli* and gut inflammation [58]. It is unknown whether the simultaneous upregulation of *Bacilli* species in the mouth and the gut environment of ASD patients represents any common causal environmental factor (such as diet), or whether overgrowth of *Bacilli* in the mouth could lead to overgrowth of *Bacilli* in the gut. Answers to these questions would help elucidate further the interactions between gut and mouth microbiomes, as well as provide insight into potential ASD pathology.

Consistent with prior reports, ASD patients demonstrated a significantly higher gut *Firmicutes*/*Bacteroidetes* ratio [59,60], which is a measure associated with inflammatory conditions such as IBD [61,62]. Overgrowth of *Proteobacteria* has been associated with diarrheal diseases, metabolic syndrome and IBD [53], and 4 out of the 6 subjects who exhibited significant *Proteobacteria* overgrowth were ASD patients. *Proteobacteria* overgrowth observed in our study is unlikely due to confounding factors: none of the six subjects were under 5 years-old (age range: 15–45), and none had used antibiotics in the past month. We also explored other putative combined phylum level relative abundance or ratios as dysbiosis markers, which all appear to be abnormal in patients with ASD.

### 4.1. Microbial Signatures Can Serve as Potential Diagnostic Markers for ASD

Although oral and gut microbiomes are distinct, we showed that analysis of both can be combined to classify ASD subjects from controls. Among the dysbiosis markers and differentially expressed taxa in the present study, three promising candidates stood out from our analysis: gut *Butyricimonas*, saliva *Parvimonas*, and gut *Firmicutes*/*Bacteroidetes* ratio. In support of our findings is the work done by Kang et al. (2013) which also reported decreased *Butyricimonas* in the gut of ASD patients as compared to controls [24]. *Butyricimonas* is prevalent in healthy individuals and produces butyrate, which has been shown to improve gut health [63]. In addition, recent work on multiple sclerosis suggests that it may play an important role in immune tolerance and prevention against disease pathogenesis and progression [64,65]. *Butyricimonas* had negative correlations with gene expression implicated in cytokine signalling molecules IFN and IL-2, and activation of receptors PPAR and RXR [64]. Given the important association between autoimmune conditions and ASD, it will be important to further explore the role of *Butyricimonas* in the pathogenesis and autoimmune manifestation of ASD patients. Another study reported thedepletion of oral *Parvimonas* in IBD patients, although this has not been reported in ASD patients [66,67].

Currently, ASD diagnosis is guided by criteria in the DSM-5, which are based solely on clinical symptoms without any objective laboratory measures. Utilizing a combination of gut and oral microbiome signatures could improve the diagnosis and screening process of ASD individuals. This could also identify subclinical or clinical subgroups of ASD patients with potential GI involvement, autoimmunity, or inflammation. Future studies should explore whether these microbiome markers can predict a patient’s response to treatment. This would be particularly useful to guide treatment with probiotics or drug options during probiotics therapy and anti-inflammatory interventions, as it could individualize treatment and improve outcomes for patients with ASD.

### 4.2. Gut and Oral Co-Occurrence Network Reveal Possible Connections between Distinct Microbial Communities

Our study is the first to co-analyze stool and oral microbiota in patients with ASD. We explored methodologies to investigate the relationship between the oral and gut microbiomes using unbiased approaches. Our analysis revealed novel co-occurrence networks within and between microbial communities that may hold diagnostic significance for ASD. Given how environmental factors (such as diet) can facilitate competitive and cooperative relationships between microbial groups [68], it is possible that such effects can span across distant communities along the digestive tract. The SparCC co-occurrence network analysis revealed an overall denser correlation network of the saliva microbiome compared to the gut. It is known that inter-individual variability of gut microbiota is higher compared with that of salivary microbiota [26], which may explain this observed difference.

Interestingly, some gut and oral taxa show evidence of co-occurrence despite the distal separation. For example, gut *Firmicutes* and saliva *Chloroflexi* showed strong correlation in the ASD population. From a diagnostic perspective, it would be pertinent to explore whether oral *Chloroflexi* can serve as a read-out for the status of gut *Firmicutes* in patients with ASD, thereby using oral microbiome as a more convenient tool to assess dysbiosis of the gut when stool samples are not readily available. More significant oral–gut co-occurrence clusters were observed at the genus level. The oral microbiome may help predict the levels of *Bifidobacteria*, *Escherichia* and *Clostridiales* genera in the gut, which all showed positive correlations with oral genera and are likely correlated with GI and/or ASD pathophysiology [12,69,70].

### 4.3. Clinical Correlates of ASD Microbiome

Despite the recognized importance of the gut microbiota in health and disease, our study is the one of the few designed to investigate the relationship between the human microbiota and medical comorbidities of ASD patients. Previously, Plaza-Diaz investigated gut microbiome in ASD patients with or without mental regression and found microbiome signatures associated with different psychiatric subtypes [32]. We analyzed gut alpha diversity, as well oral and gut phylum and genus levels of relative abundance in the context of three common co-occurring medical conditions affecting the ASD individuals: allergies, abdominal pain and poor dietary habits.

We found ASD patients tend to have more unhealthy and restricted dietary habits. This is consistent with previous studies, showing that up to 79% of children with ASD suffer from feeding-related difficulties or nutritional challenges [71] and strong preference for nutrient-poor foods [72]. Given the correlation between severity of poor dietary habits and relative abundances of gut microbiome biomarkers, it is conceivable that the unhealthy dietary habits may be driving gut dysbiosis [73].

Second, we detected a significantly higher prevalence of allergies in ASD patients with than those without. Gut *Proteobacteria* overgrowth is also over-represented in ASD patients and its relative abundance is positively correlated with allergy status. Overgrowth of *Proteobacteria* has been implicated in autoimmune disorders such as IBD [74]. This is opposite to the trend of *Bacteroidetes*, a marker for healthy flora. We also report a negative association between oral *SR1* numbers and allergy status, but this association is only present in ASD patients and not healthy controls.

Little is known about the connection between allergies and autism. In a recent, large population-based, cross-sectional study of data provided by the National Health Interview Survey (NHIS) from 1997–2016, Xu et al. found that children with ASD were more likely to have a food allergy (11.25% versus 4.25%), respiratory allergy (18.73% versus 12.08%), and skin allergy (16.81% versus 9.84%) than neurotypical children. Further, the odds ratio of ASD among children with a food allergy is nearly triple the ratio of ASD among those without a food allergy [75].

The “bi-directional” association between allergies and ASD raises the following questions: (1) whether these dysbiosis markers are simply associated with allergy or whether an abnormal microbiome is involved in the pathogenesis of allergy in ASD, (2) if a pathogenic mechanism could be established, whether ASD patients are more vulnerable to it than neurotypical individuals, and (3) whether there are common underlying mechanisms, potentially involving the dysregulation of the immune system and gut and oral microbiota, that could induce the development of both allergy and ASD. Future studies using animal models, immunology markers, genomics and metabolomics approaches are needed to elucidate the mechanisms of possible causal relationships.

In analyzing the relationship between microbiota and GI pathology, we found significantly higher levels of gut *Roseburia* in subjects without abdominal pain. The genus *Roseburia* consists of obligate Gram-positive anaerobic commensal bacteria that affect one’s health in many ways. These bacteria produce short-chain fatty acids such as butyrate, affect colonic motility, maintain the immune response, and contribute anti-inflammatory factors to their environments [76]. Although previous studies have linked *Roseburia* abundance to some disease states such as irritable bowel syndrome and IBD [77], certain species in the genus likely play a positive role in GI health. One recent study found that treatment with the *Roseburia hominis* bacterium provided protection against dextran sulfate sodium (DSS)-induced colitis due to its immunomodulatory properties [78].

Interestingly, the oral genus *Porphyromonas* is significantly more abundant in subjects with abdominal pain. Many members of this genus have been associated with periodontal diseases [79]. The most well-characterized species, *Porphyromonas gingivalis*, has been linked to systemic diseases including upper GI tract inflammation and cancer due to upregulation of systemic cytokine release [80]. Further investigations should consider the mechanistic roles these genera could play in abdominal pain, and whether these gut and oral genera can serve as markers for the diagnosis and treatment monitoring of abdominal symptoms in patients with ASD.

The correlation between abdominal pain status and differential expression of bacterial genera differs between the ASD and control groups. Previously, Strati et al. found that constipation status is correlated with different amounts of bacterial taxa depending on whether an individual has ASD or not [23]. Notably, *Bacteroides* is one genus that shows the most prominent differential patterns: whereas *Bacteroides* appears to be protective against abdominal pain in ASD patients (higher levels are associated with no abdominal pain), the association is the opposite in controls. *Bacteroides* genus harbor species that can have either positive or negative effects on GI health. Some *Bacteroides* species synthesize lipopolysaccharide, an important bacterial virulence factor, and can cause diseases such as GI infection and septicemia in children. Many other *Bacteroides* species can be healthy commensals [81]. A recent meta-analysis concluded that a lower level of *Bacteroides* in the gut microbiota is associated with IBD [82], and functional analysis showed that *Bacteroides* expresses polysaccharide A, which can induce regulatory T-cell growth and cytokine expression to protect against colitis [83]. It is possible that ASD patients may be more prone to positive effects of *Bacteroides* than control subjects, potentially through the action of bacterial metabolites and the gut–brain axis [13]. This is supported by a mouse study which found that administration of *Bacteroides fragilis* corrects gut permeability, alters microbial composition and ameliorates ASD-related defects [13]. The ASD mice also display an altered serum metabolomic profile, and *B. fragilis* modulates levels of several metabolites. Further species-level analysis with higher 16S rRNA gene amplicon sequencing resolution and functional studies could elucidate the roles of different Bacteroides species on abdominal pain in ASD subjects. Future studies should also investigate the relationships between abdominal pain, *Bacteroides* abundance, and the severity of ASD symptoms.

## 5. Conclusions

In conclusion, our study is the first to use a first degree-relative matched design combined with high fidelity 16S rRNA gene amplicon sequencing technology to characterize the microbiome of patients with ASD compared to neurotypical individuals. To our knowledge, this study is the first to co-analyze the oral and gut microbiomes in patients with ASD, as well as explore the relationship between the two microbial communities and clinical indices. This study identified distinct features of gut and salivary microbiota that differ between individuals with and without an ASD diagnosis. The diagnosis of ASD currently relies on psychological testing with potential high subjectivity and inconsistencies. We suggest improvement of current diagnostic approaches based on gut and oral microbial signatures and co-occurrence networks. Given the emerging role that the human microbiome plays in systemic diseases, we hope that these analyses will provide clues for developing microbial markers for diagnosing ASD and comorbid conditions, and to guide treatment. In particular, ASD patients have disproportional gastrointestinal symptoms compared to neurotypical individuals. Therefore, developing “gut microbiome markers” is particularly important for monitoring GI health or guiding interventions of the gut. For example, these preliminary results can serve as a starting point to test whether changing the microbiome (e.g., with probiotics) would improve co-morbid conditions in patients with ASD and further modify the core and GI symptoms of ASD.

The explorations of causal relationships between microbiomes, ASD status and co-morbidities await future investigations. Further research could explore metabolomics profiles to characterize microbiome-related inflammatory factors and metabolites in the oral and gut cavity such as interleukins and short-chain fatty acids. Other areas of future study should include exploring the role of microbiota in inflammatory conditions such as allergy and autoimmunity, investigating their genetic and/or epigenetic linkage, researching mechanism of the gut–brain axis and relevant neural circuits, and ultimately inquiring more about the pathogenesis of ASD. These indices and studies will improve the algorithm for ASD screening, diagnosis, and treatment monitoring in the future.

Limitations of the current study include: (1) The use of both sibling and parental controls, where age could contribute to the large inter-individual variability. Future studies should focus on only age-matched sibling controls, if possible. (2) The small sample size, which likely contributed to high FDR in the majority of our analyses and the difficulty in distinguishing true differences from noise. Verification of our findings with a larger cohort is required. The current study was not sufficiently powered for detecting clinically relevant biomarkers. However, with the methodologies in hand, we will be able to expand the study to develop clinically biomarkers in the future. That being said, even with the relatively small sample size, we were able to find biomarkers that have withstood rigorous statistical testing and adjustment. (3) Our genus level differential expression patterns showed discrepancies from previous reports that used neurotypical controls [12,23], but this likely reflects the differences in study design [12]. For example, we did not detect changes in *Prevotella*, *Bacteroides*, *Clostridium* cluster I/II, or *Lactobacillus*, which have been reported by some studies to be differentially expressed between ASD and control groups [12], but previous studies using sibling-matched designs also did not detect these differences [84,85,86].

## Figures and Tables

**Figure 1 nutrients-11-02128-f001:**
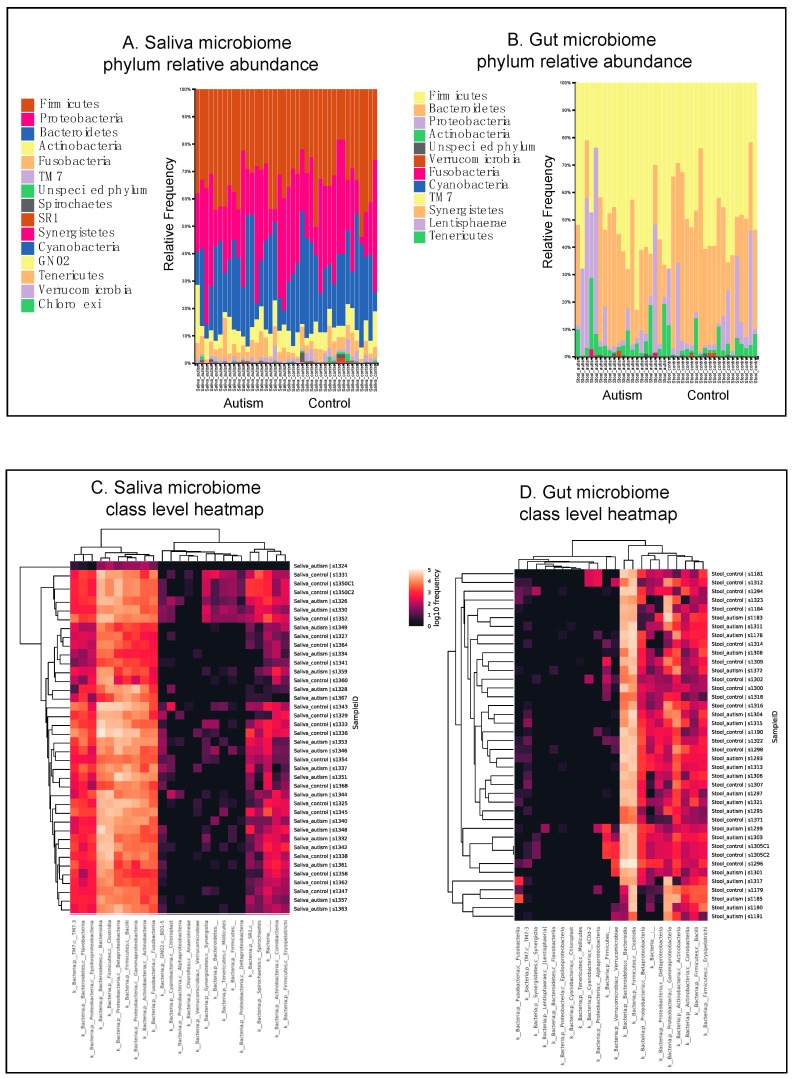
Bar plots of bacterial phylum-level relative abundances of the salivary (**A**) and gut (**B**) microbiomes. Each bar represents one subject. (**C**) Salivary microbiome class-level heatmap expression profile. (**D**) Gut microbiome class-level heatmap expression profile.

**Figure 2 nutrients-11-02128-f002:**
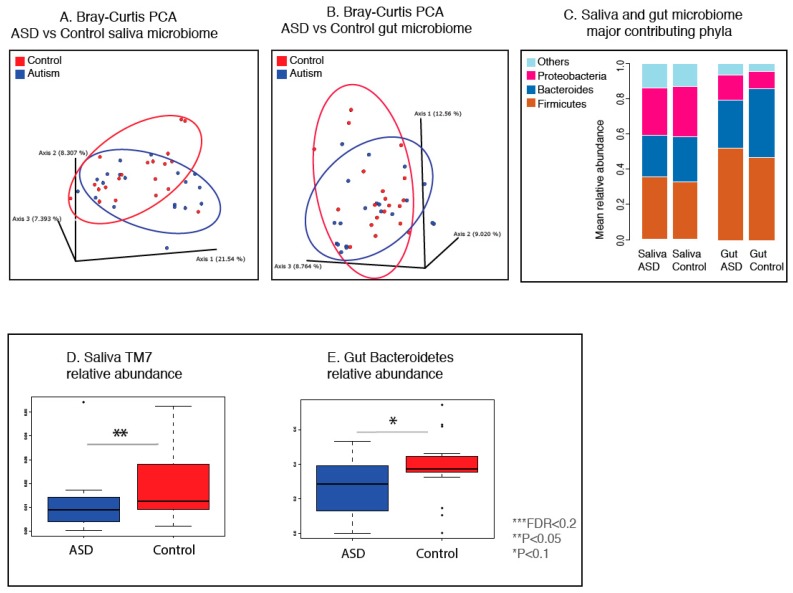
PCA of bacterial beta diversity of saliva (**A**) and gut (**B**) microbiomes based on the Bray–Curtis dissimilarity for ASD and neurotypical subjects. ASD and neurotypical subjects are colored in blue and red, respectively. (**C**) The major contributing phyla of gut and oral microbiome, in ASD and control subjects. The values used to compose the figures represent group mean relative abundances. (**D**,**E**) Box plots depicting relative abundances of the most differentially abundant salivary or gut bacterial phyla between patients with ASD and control subjects. Single asterisk indicates *p* < 0.1 with adjusted FDR > 0.2; double asterisk indicates *p* < 0.05 with adjusted FDR > 0.2, triple asterisk indicates *p* < 0.05 and adjusted FDR < 0.2, Kruskal–Wallis test.

**Figure 3 nutrients-11-02128-f003:**
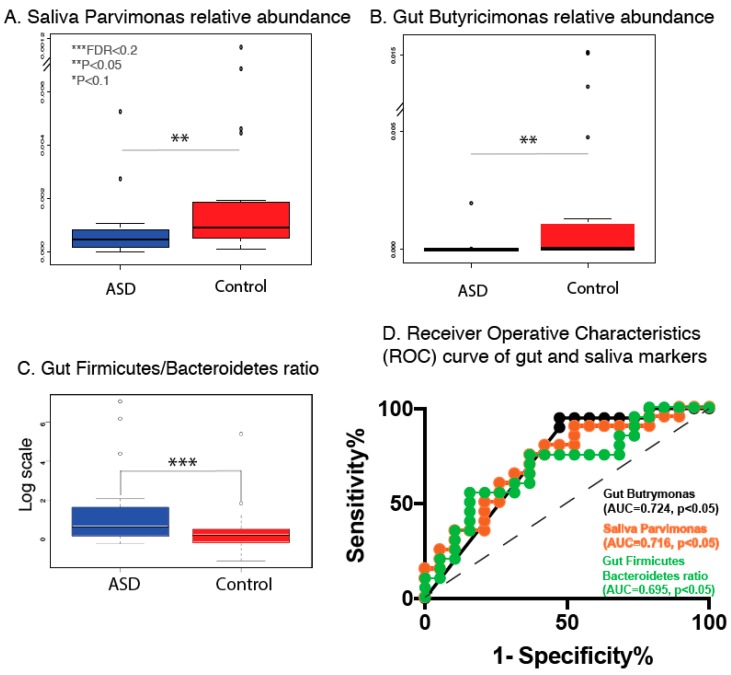
(**A**,**B**) Box plot representations of the relative abundances of differentially abundant salivary or gut bacterial genera in patients with Autism Spectrum Disorder (ASD) and control subjects. (**C**) Box plots representation of gut phylum-level dysbiosis marker *Firmicutes/Bacteroidetes* ratio, in patients with ASD and control subjects. ASD and neurotypical subjects are colored in blue and red, respectively. Single asterisk indicates *p* < 0.1 with adjusted FDR > 0.2; double asterisk indicates *p* < 0.05 with adjusted FDR > 0.2, triple asterisk indicates *p* < 0.05 and adjusted FDR < 0.2, Kruskal–Wallis test. (**D**) receiver operator characteristics (ROC) curve of the 3 differentially abundant gut or oral genera and dysbiosis markers that have the highest area under the curve (AUC), and *p* < 0.05 based on two-sided Z-test for ROC.

**Figure 4 nutrients-11-02128-f004:**
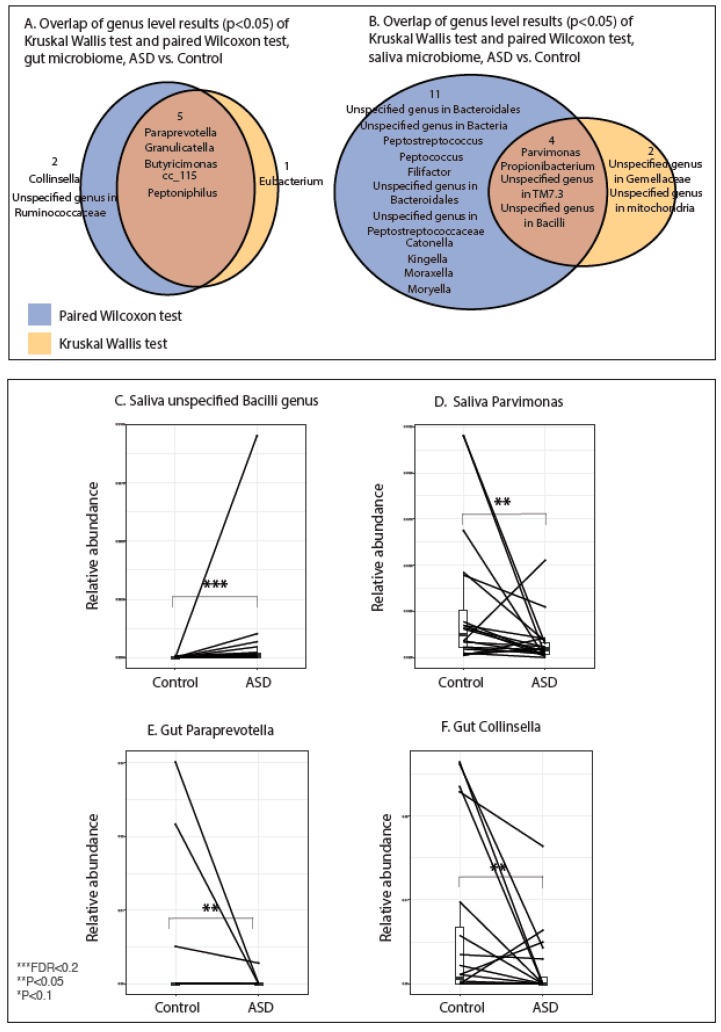
(**A**,**B**) Overlap of differentially abundant gut or salivary genera based on Kruskal–Wallis test and paired Wilcoxon test. Results are for taxa with unadjusted *p* < 0.05. (**C**,**D**) Paired-test representation of the relative abundances of top most differentially abundant salivary bacterial genera between ASD patient–family member control pairs. (**E**,**F**) Paired-test representation of the relative abundances of top most differentially abundant gut bacterial genera between ASD patient–family member control pairs. Single asterisk indicates *p* < 0.1 with adjusted FDR > 0.2; double asterisk indicates *p* < 0.05 with adjusted FDR > 0.2, triple asterisk indicates *p* < 0.05 and adjusted FDR < 0.2, Wilcoxon’s paired test.

**Figure 5 nutrients-11-02128-f005:**
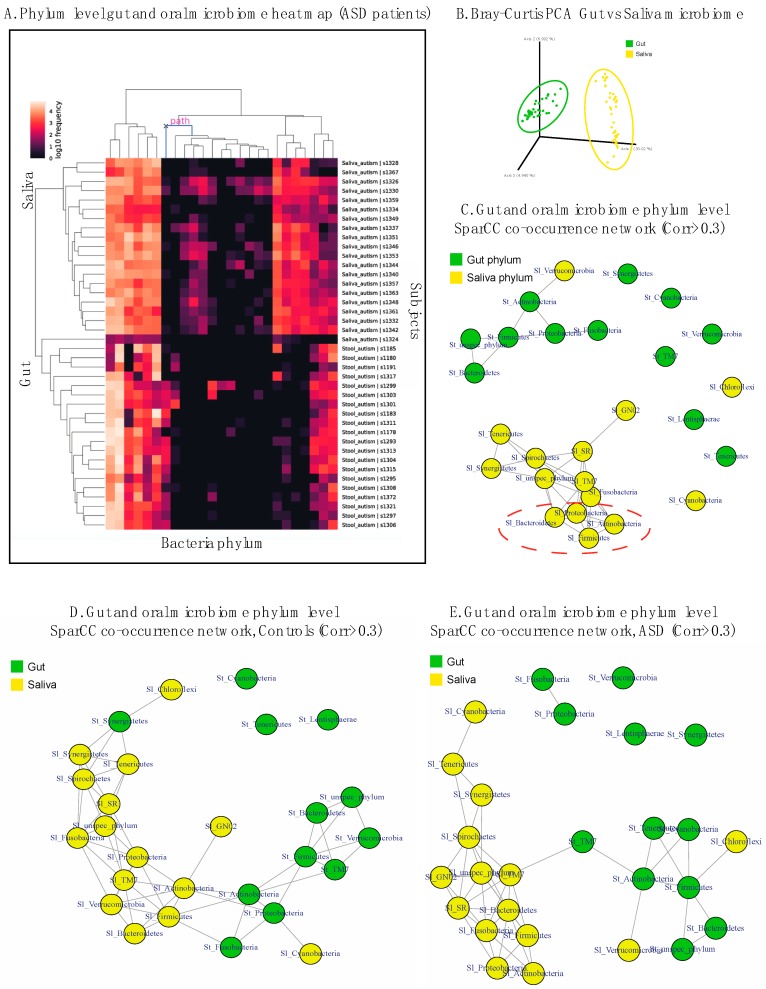
(**A**) Phylum-level heat map expression profiles of gut and oral microbiomes in ASD patients. (**B**) PCA of bacterial beta diversity based on Bray–Curtis dissimilarity for saliva and gut (all subjects are represented). Saliva and gut microbiome are colored in yellow and green, respectively. (**C**–**E**) Gut and oral microbiome phylum level co-occurrence network using the Sparse Correlations for Compositional data (SparCC) method with a correlation cut-off >0.3 ((**C**) all subjects, (**D**) control only, (**E**) ASD only). Each node represents a saliva (Sl) or stool (St) phylum, and saliva and stool microbiomes are colored in yellow and green, respectively. The dotted red circle highlights a co-occurrence cluster with the greatest inter-nodal correlations.

**Figure 6 nutrients-11-02128-f006:**
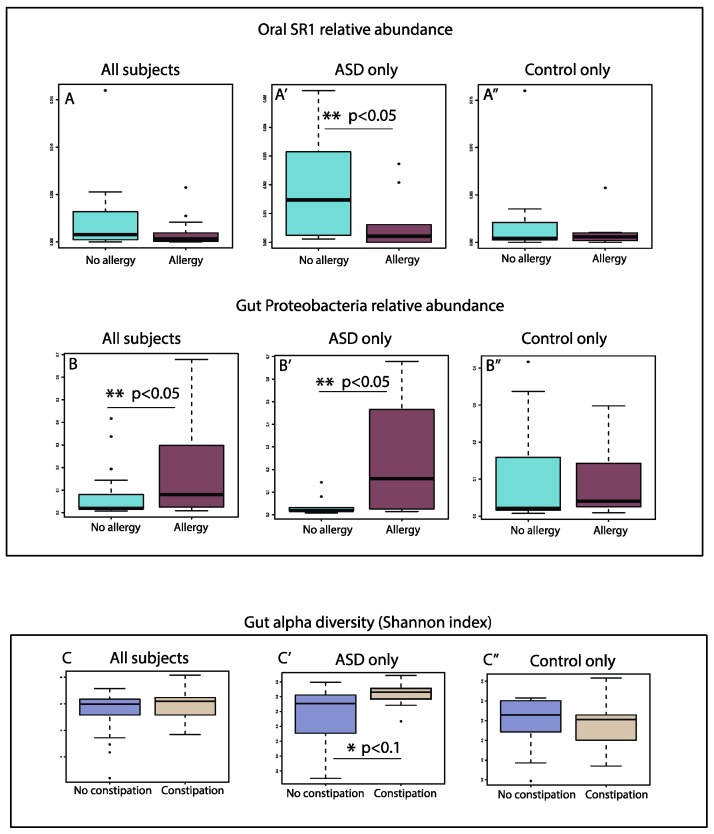
Box plot representation of the relative abundances of oral (**A**–**A”**) and gut (**B**–**B”**) bacterial phyla correlating with the allergy status of the subjects enrolled in this study. (**A**) Oral SR1 relative abundance in all subjects with no allergy and those with allergy; (**A’**) Oral SR1 relative abundance in ASD subjects with no allergy and patients with allergy; (**A”**) Oral SR1 relative abundance in neurotypical subjects with no allergy and neurotypical subjects with allergy. (**B**) Gut *Proteobacteria* relative abundance in all subjects with no allergy and those with allergy; (**B’**) Gut *Proteobacteria* relative abundance in ASD subjects with no allergy and patients with allergy; (**B”**) Gut *Proteobacteria* relative abundance in neurotypical subjects with no allergy and neurotypical subjects with allergy. Box plot representation of the gut alpha diversity (Shannon index) that correlated with the allergy status of the subjects enrolled in this study. (**C**) Gut alpha diversity in all subjects with no constipation and those with constipation; (**C’**) Gut alpha diversity in ASD subjects with no constipation and patients with constipation; (**C”**) Gut alpha diversity in neurotypical subjects with no constipation and neurotypical subjects with constipation. Single asterisk indicates *p* < 0.1 with adjusted FDR > 0.2; double asterisk indicates *p* < 0.05 with adjusted FDR > 0.2, triple asterisk indicates *p* < 0.05 and adjusted FDR < 0.2, Kruskal–Wallis test.

**Figure 7 nutrients-11-02128-f007:**
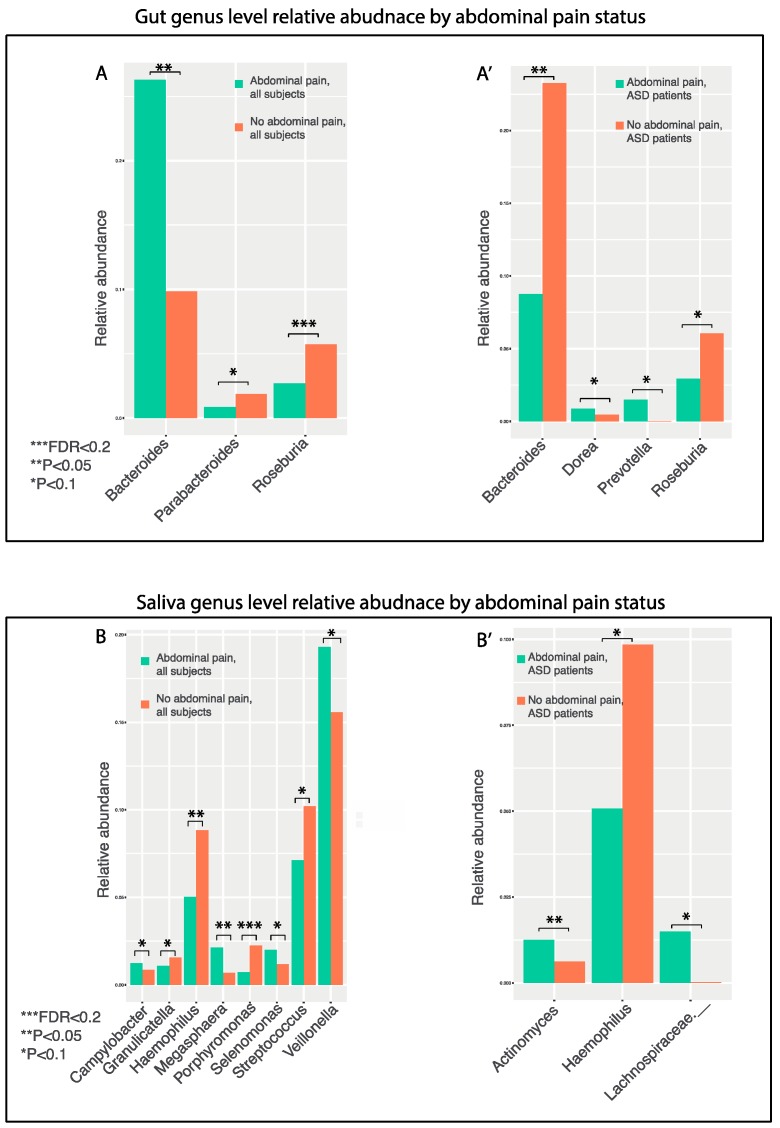
Bar plot representation of the relative abundances of gut (**A**–**A’**) and oral (**B**–**B’**) bacterial genera correlating with the abdominal status of the subjects enrolled in this study. (**A**) The most differentially abundant gut genera in all subjects with no abdominal pain and those with abdominal pain; (**A’**) The most differentially abundant gut genera in ASD patients with no abdominal pain and patients with abdominal pain. (**B**) The most differentially abundant oral genera in all subjects with no abdominal pain and those with abdominal pain; (**B’**) The most differentially abundant oral genera in ASD patients with no abdominal pain and patients with abdominal pain. Single asterisk indicates *p* < 0.1; double asterisk indicates *p* < 0.05, Kruskal–Wallis test.

**Figure 8 nutrients-11-02128-f008:**
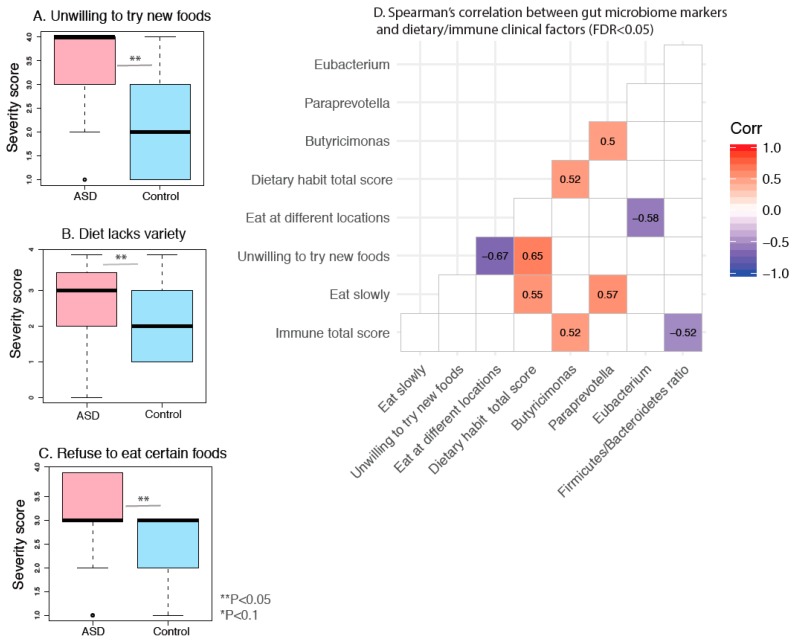
Box plot representation of abnormal dietary habit severity scores in ASD and control subjects. (**A**) Unwilling to try new foods. (**B**) Diet lacks variety. (**C**) Refuse to eat certain foods. Single asterisk indicates *p* < 0.1; double asterisk indicates *p* < 0.05, Mann–Whitney U test. (**D**) Spearman’s correlation matrix between habit scores, allergy/autoimmunity scores, gastrointestinal severity indices (GSI) total score, and selected ASD gut microbiome markers in patients with ASD (results with FDR < 0.05 were shown).

**Table 1 nutrients-11-02128-t001:** Characteristics of study participants and microbiome lifestyle factors.

	*Autistic*	*Neurotypical*
Subjects	20	19
Age (1st–3rd quartile)	15 (13–18)	29 (11–50)
Gender (n)		
Female	25% (5)	58% (11)
Male	75% (15)	42% (8)
Neighborhood in last 5 years (n)		
Cities	10% (2)	11% (2)
Suburbs	90% (18)	89% (17)
Countryside	0% (0)	0% (0)
Pets (n)		
Yes	10% (2)	11% (2)
No	85% (17)	84% (16)
n/a	5% (1)	5% (1)
Antibiotics/Probiotics (n)		
Prior use in the last month	35% (7)	21% (4)
No use in the last month	65% (13)	79% (15)
Constipation (n)		
5 or more stools/week	55% (11)	63% (12)
3–4 stools/week	30% (6)	26% (5)
0–2 stools/week	10% (2)	5% (1)
n/a	5% (1)	5% (1)
Abdominal tenderness during exam (n)		
Yes	0% (0)	0% (0)
No	95% (19)	95% (18)
n/a	5% (1)	5% (1)
Allergies (n)		
Yes	60% (12)	37% (7)
No	40% (8)	63% (12)
Drink alcohol (n)		
Yes	0% (0)	11% (2)
No	95% (19)	84% (16)
n/a	5% (1)	5% (1)
Recreational drugs (n)		
Yes	0% (0)	0% (0)
No	95% (19)	19% (18)
n/a	5% (1)	5% (1)
Tobacco products (n)		
Yes	0% (0)	0% (0)
No	95% (19)	95% (18)
n/a	5% (1)	5% (1)
First 6 months of life		
Breast Fed	70% (14)	74% (14)
Bottle Fed	15% (3)	5% (1)
Both	25% (5)	16% (3)
n/a	5% (1)	5% (1)
Picky Eater		
Yes	20% (4)	11% (2)
No	80% (16)	84% (16)
n/a	0% (0)	5% (1)
Servings of vegetables and fruits per day (n)		
Less than three	65% (13)	74% (14)
Three	30% (6)	21% (4)
More than three	5% (1)	5% (1)

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
