# Peer review of "New and Preliminary Evidence on Altered Oral and Gut Microbiota in Individuals with Autism Spectrum Disorder (ASD): Implications for ASD Diagnosis and Subtyping Based on Microbial Biomarkers"

_nutrients, 2019, doi:10.3390/nu11092128_

Round 1
Reviewer 1 Report
This cross-sectional study primarily aimed to characterize the oral and gut microbiome in those with ASD and compare with neurotypical controls, and secondarily aimed to correlate oral and gut microbiome with other factors such as diet and GI issues. The goal was to identify microbiome markers to monitor responses to a subsequent clinical trial using probiotic supplementation. The methodology and analysis are sound and the results are interesting, yet not sure how sensitive will be for detecting changes that may be clinically relevant. Overall, well-done study and I recommend to accept with minor edits.
Author Response
Response: We appreciate the reviewer’s insightful comments. This study is a pilot study with the goal of mapping out methodologies to select microbiome biomarkers for ASD research, with the hope of developing clinically relevant biomarkers in the future. Therefore, the current study in itself is not powered for detecting clinically relevant biomarkers yet. With the methodologies in hand, we will expand our study to include more subjects to develop clinically relevant biomarkers. That being said, even with the small sample size, we are able to find biomarkers that have withstood rigorous statistical testing and adjustment. We will be testing the relevance of these markers in the upcoming study with bigger sample size, as well as in our ongoing clinical trial. We have included a discussion of this point in our revised manuscript. In addition, we have edited grammatical errors as suggested.
Reviewer 2 Report
Thank you very much for the opportunity to review this work. This well-designed and well-implemented initial work will add a great deal to emerging science and future research. I see no revisions necessary in current content though there are several very minor grammatical/spelling concerns to address throughout the paper.Reviewer 3 Report
Interesting and important study.
But the major question is whether the changes in microbiota observed is the cause or effect of the disease.
Is there any disease that is not associated with changes in gut microbiota as of today?!?
The major issue with this kind of studies is the mechanisms of action of altered gut/oral and other types of microbiota (or for that matter any type of microbiota) in the pathobiology of the disease.
In all these studies, the interaction(s) between microbiota, their metabolites, gut/oral cells, microbiota and T cells and macrophages, local stem cells and microbiota, etc., is missing. Observing a change in microbiota in a disease (inclduing ASD) is just an association. It dos not mean that they are involved in the disease itself.
In a disease like ASD where genetic, environmental, neurones, neurotransmitters and maternal and paternal factors have a role, it is extremely difficult to suggest that microbiota only have a role.
Nevertheless, it is an good attempt on the part of the authors.
Author Response
Response: We appreciate the reviewer’s insightful comments. We would like to address the two major issues raised by the reviewer.
1) The major issue with this kind of studies is the mechanisms of action of altered gut/oral and other types of microbiota (or for that matter any type of microbiota) in the pathobiology of the disease.
We are aware of this major limitation of the study, which is why the goal of our study is not to investigate pathophysiology of the disease, but to identify microbiome markers to monitor responses to a subsequent clinical trial. Although association is by no means causal, it can be a powerful factor to consider if it can help guide/predict treatment. This is equivalent to using fever curve to monitor treatment of sepsis: although fever itself is not a pathogenic factor but rather a by-product of infection/inflammation, it is valuable as a marker of treatment response. In the subsequent clinical trial, we have included mechanistic investigation of the gut brain axis as suggested by the reviewer.
2) As to why we care about microbiome in ASD when microbiome is associated with so many diseases: we would like to argue that ASD is a little different: unlike Alzheimer’s disease or diabetes (just as two examples), ASD patients have disproportional gastrointestinal symptoms compared to neurotypical individuals, which is why developing “gut microbiome markers” is particularly important in this population for monitoring gastrointestinal health/guiding interventions of the gut, as opposed to patients with many other diseases.